# Endoscopic Ultrasound in Pancreatic Duct Anomalies

**DOI:** 10.3390/diagnostics13193129

**Published:** 2023-10-05

**Authors:** Abhirup Chatterjee, Surinder Singh Rana

**Affiliations:** Department of Gastroenterology, Post Graduate Institute of Medical Education and Research (PGIMER), Sector 12, Chandigarh 160012, India; drabhirup@gmail.com

**Keywords:** pancreas divisum, endosonography, annular pancreas, magnetic resonance cholangiopancreatography

## Abstract

Embryological development of the pancreas is a complex phenomenon and, therefore, it can have multiple developmental aberrations. Fortunately, the majority of these pancreatic ductal anomalies are asymptomatic with no clinical relevance and are incidentally detected during diagnostic cross-sectional imaging or endoscopic retrograde cholangiopancreatography (ERCP) or autopsy. Occasionally, pancreatic duct anomalies can result in symptoms like abdominal pain or recurrent pancreatitis. Also, an accurate pre-operative diagnosis of ductal anomalies can prevent inadvertent duct injury during surgery. Conventionally, ERCP had been used for an accurate diagnosis of pancreatic duct anomalies. However, because it is invasive and associated with a risk of pancreatitis, it has been replaced with magnetic resonance cholangiopancreatography (MRCP). MRCP has demonstrated high sensitivity and specificity for the diagnosis of ductal anomalies, which can be further improved with the use of secretin-enhanced MRCP. Endoscopic ultrasound (EUS) is a new diagnostic and interventional tool in the armamentarium of endoscopists and has demonstrated promising results in the detection of pancreatic duct variations and anomalies. Along with the visualization of the course and configuration of the pancreatic duct, EUS can also visualize changes in the pancreatic parenchyma, thereby helping with an early diagnosis of any co-existent pancreatic disease. Absence of the stack sign and crossed duct sign are important EUS features to diagnose pancreas divisum. EUS can also help with the diagnosis of other congenital ductal anomalies like annular pancreas, ansa pancreatica, and anomalous pancreaticobiliary union, although the published experience is limited.

## 1. Introduction

Aberrations during embryological development of the pancreas can result in various anatomic variations of the pancreas and pancreatic duct including variations in either the course of the pancreatic duct, such as the descending, vertical, loop, or sigmoid course, or configuration of the pancreatic duct, such as bifid configuration with dominant duct of Wirsung, dominant duct of Santorini without divisum, absent duct of Santorini and ansa pancreatica, anomalous pancreaticobiliary ductal junction, pancreas divisum, annular pancreas, ectopic pancreas, pancreatic agenesis and hypoplasia of the dorsal pancreas, and accessory pancreas P [1]. The majority of these developmental anomalies do not have clinical significance and are incidentally detected during cross-sectional imaging or endoscopic retrograde cholangiopancreatography (ERCP). Improvements in cross-sectional imaging techniques, especially the development of high-resolution magnetic resonance cholangiopancreatography (MRCP), have improved our diagnostic ability to non-invasively diagnose pancreatic ductal as well as parenchymal anomalies [2]. Endoscopic ultrasound (EUS) also provides high-resolution images of the pancreas, as well as the pancreatic duct, and has excellent accuracy for the diagnosis of various pancreatic parenchymal and ductal anomalies [3]. The technical details and the accuracy of EUS in diagnosing congenital pancreatic anomalies have not been reviewed and, therefore, in this review article, we will focus on the role of EUS in the diagnosis of various pancreatic duct anomalies. For this review, we performed an electronic search in PubMed using the following medical subject heading terms: endoscopic ultrasound; congenital pancreatic anomalies; pancreas divisum; annular pancreas; ansa pancreatica; and anomalous pancreaticobiliary union. The search included all types of articles published in the English language until July 2023.

## 2. Embryological Development of the Pancreas

Proper understanding of various congenital pancreatic anomalies requires an in-depth understanding of its embryological development. The development of the pancreas during embryogenesis starts during the fifth week with the development of two buds: a larger dorsal bud, and the smaller ventral bud, from the second part of the duodenum. The ventral bud arises from the ventral outpouching of the gut that also forms the liver, cystic duct, gall bladder, bile duct, and, therefore, the common bile duct (CBD), and the ventral pancreatic duct share common embryonic origin. The dorsal bud, after arising from the endoderm, starts growing in a left-lateral direction, and by sixth week, it enters the dorsal mesentery to become intraperitoneal. The smaller ventral bud arises from the hepatic diverticulum and under the effect of a complex interaction between epithelium and the mesenchyme, the ventral bud migrates with the bile duct and rotates clockwise around the duodenum to fuse with the dorsal bud by the seventh embryonic week [4,5,6].

Afterwards, due to further growth of the foregut, the ventral bud and bile duct rotate counter-clockwise along the duodenum and take a dorsal position, and the pancreas grows as one organ with the joining of the bile duct with the pancreatic duct. The pancreatic ductal system starts developing once the ventral and dorsal part fuse together, side-by-side. The larger dorsal bud ultimately leads to the formation of the neck, body, and tail of the pancreas, whereas the smaller ventral bud forms the head and uncinate process. The duct of the dorsal duct, also known as the duct of Santorini (the accessory pancreatic duct), drains into the minor papilla. Similarly, a new duct connects the distal part of the dorsal duct with the shorter duct of the ventral pancreas leading to the formation of the main pancreatic duct (the duct of Wirsung), which drains into the major papilla. The accessory pancreatic duct joins the main pancreatic duct about 1–2 cm proximal to the ventral duct or at the distal end of the ventral duct. The part of the accessory pancreatic duct that is distal to the union of the two ducts becomes stenotic and may become obliterated; thus, functionally playing a minor role in drainage of the pancreatic exocrine secretion, which preferentially flows through the main pancreatic duct [6,7,8].

## 3. Types of Pancreatic Duct Anomalies

Physiologically, developmental anomalies of the pancreas can be broadly divided into (i) fusion anomalies, (ii) migration anomalies, and (iii) duplication anomalies (Table 1). The pancreatic duct variations can also be divided on the basis of its course, configuration, anomalous union, and pancreas divisum (Table 2).

## 4. Endoscopic Ultrasound (EUS) in Pancreatic Diseases

EUS has revolutionized the diagnosis and management of pancreatic diseases. The superiority of EUS over other imaging modalities is due to its ability to provide high-resolution images of both the pancreatic parenchyma and the pancreatic duct. EUS is the investigation of choice for etiological evaluation in idiopathic pancreatitis and for early diagnosis of chronic pancreatitis [9,10,11]. By virtue of its ability to provide high-resolution images of the pancreatic duct, EUS can also help with an accurate diagnosis of various congenital pancreatic ductal anomalies. Along with the visualization of the course and configuration of the pancreatic duct, EUS can also visualize changes in the pancreatic parenchyma, thereby helping with the early diagnosis of any co-existent pancreatic disease.

## 5. Pancreas Divisum

Pancreas divisum is the most common type of anatomical anomaly of the pancreatic duct, which results from failed or abnormal fusion of the ventral and the dorsal pancreatic duct. This results in preferential drainage of the pancreatic secretion by the duct of Santorini into the minor papilla covering the body and tail region of the pancreas, whereas the duct of Wirsung plays a marginal role in drainage into the major papilla [12]. The frequency of pancreas divisum has been variably reported with the autopsy series reporting frequency varying between 4.4 and 12% and ERCP studies reporting the frequency varying between 0.3–8% [13].

The clinical significance of pancreas divisum is controversial because of conflicting evidence. The majority of patients with pancreas divisum have no pancreatic symptoms throughout their life and, therefore, it is suggested that this anatomical variant has little or no clinical significance [14,15]. However, few studies have demonstrated an increased frequency of pancreas divisum in patients with idiopathic pancreatitis, suggesting its pathogenic role [13,16,17]. The relief of pain following minor papilla sphincterotomy suggests that the minor papilla orifice in these symptomatic patients is critically narrowed, resulting in the impaired outflow of pancreatic juice and the increased intraductal pressure, resulting in abdominal pain and pancreatitis [18]. However, this “obstructive pancreatopathy” concept has been challenged by recent genetic studies that have suggested that the cystic fibrosis trans-membrane conductance regulator (CFTR), or other unidentified genetic mutations, predisposes pancreas divisum patients to recurrent acute or chronic pancreatitis rather than impaired pancreatic drainage [19,20]. The debate about its exact pathogenic role continues, but a few patients with symptomatic pancreas divisum do get relief following successful endotherapy.

Pancreas divisum has been classified into three types based upon the anatomy of the ductal system. Type 1 is when the dorsal and ventral system are completely separated; type 2 is when the ventral duct is completely absent; and type 3 is when there is a rudimentary duct connecting the dorsal and the ventral ductal system. Complete divisum implies a lack of fusion between the dorsal and ventral duct (Type 1); whereas, in incomplete divisum, there is a rudimentary channel between the dorsal and ventral duct, as in type 3. Clinically, complete and incomplete divisum have similar behaviors [21,22].

Pancreas divisum can be associated with cystic dilatation of the dorsal pancreatic duct at the junction with the minor papilla. This is known as a santorinocoele and is hypothesized to occur due to a stenosed minor papilla along with the weakness of the ductal wall. Apart from Santorinocoele, a dilated dorsal duct can be seen in chronic pancreatitis, intraductal papillary mucinous neoplasm (IPMN), and ductal adenocarcinoma [23,24,25].

Pancreatic pseudodivisum or “false pancreas divisum” is an entity described in the literature where there is an acquired defect observed on the pancreatogram, resulting from a blockade of the ventral pancreatic duct by malignancy, stricture (as in chronic pancreatitis), or mucous plug, in the presence of normal fusion between the ventral and dorsal duct. Usually, it is seen in older males with a prolonged history of alcohol use, unlike pancreas divisum, which is seen in young females [26].

## 6. Endoscopic Ultrasound Features of Pancreas Divisum: Radial and Linear EUS

Pancreas divisum can be diagnosed by radial and linear echoendoscope. EUS features that suggest the presence of pancreas divisum include the absence, of the stack sign; presence of the crossed duct sign, and an inability to trace the main pancreatic duct through the body, starting from the major duodenal papilla (Table 3) [12]. Here, we will first discuss, in brief, the conventional techniques of EUS and the findings with respect to the pancreatic duct, followed by the findings that help to provide a diagnosis of pancreas divisum.

In routine endosonographic examinations, all parts of the pancreas are comprehensively examined when seen from three stations: the apex of the duodenal bulb, papilla, and distal to the papilla. Among these, the best position is the apex of the duodenal bulb as it brings the major portion of the head of the pancreas, distal common bile duct, and portal vein in the same frame. For positioning, the EUS scope is advanced along the greater curvature of the stomach and when pylorus is visible, the tip of the scope is negotiated through it, followed by air insufflation of the duodenal bulb. This is followed by gentle downward deflection of the tip of the scope, making the duodenal bulb visible. Doppler imaging helps differentiating bile duct from the arteries (the hepatic artery and gastroduodenal artery) and the portal vein. At this point, the endosonologist gets the view of the distal CBD, pancreatic duct, and portal vein in a single frame in which one structure appears to lie on top of the other, and this is called the stack sign (Figure 1). Absence of the stack sign suggests the possibility of pancreas divisum (Figure 2). As these structures do not lie in the same plane, various maneuvers like clockwise and counter-clockwise rotation and right and left torque are required for a detailed examination of these structures [12]. Although this sign is conventionally described in radial EUS, linear EUS can also detect similar anatomical configuration, although there are some subtle differences in linear EUS. In linear EUS, usually, the “stack” consists of the distal CBD and pancreatic duct, which are seen on a parallel axis (the portal vein is not seen). However, the superior mesenteric vein (SMV) or artery (SMA) can be seen on linear EUS on a different axis, once a clockwise rotation is performed, and the origin of the portal vein from the SMV can also be easily demonstrated (Figure 3) [27].

The crossed duct sign, which is also described in MRCP, has been conventionally described in radial EUS. In patients with pancreas divisum, while examining from the duodenal bulb and gradually withdrawing the scope towards the minor papilla, the dorsal duct can be seen crossing the CBD when viewed from the duodenal bulb [27]. In linear EUS, the course of the pancreatic duct can be traced from the papilla and duct of Wirsung, and duct of Santorini can be traced backwards from the major and minor duodenal papilla, respectively. In pancreas divisum, the minor papilla can easily be identified (may appear dilated at times) due to the prominent duct of Santorini. The identification of a more dilated dorsal duct compared to the ventral duct is another ancillary finding which reinforces the diagnosis, especially after secretin injection.

In about three fourths of a normal population, a distinction between the dorsal and ventral pancreas can be identified via EUS as the ventral part (often termed as the ventral analge) appears hypoechoic and the dorsal part appears hyperechoic (brighter), with a characteristic border at the interface. This is also known as the ventral–dorsal transition. In pancreas divisum, the ventral pancreatic duct does not cross this border as there is fusion failure. Thus, tracing the pancreatic duct crossing this transition border rules out pancreas divisum [27]. The diagnosis of pancreas divisum can be excluded if the pancreatic duct can be traced from the major papilla to the pancreatic body or dipping at the genu towards the major papilla when seen from the stomach (Figure 4) [28].

On linear EUS, pancreas divisum can be demonstrated from three stations with different findings as follows: (I) When the scope is in the duodenal bulb, the dorsal pancreatic duct can be visualized closer to the duodenal wall, above the CBD (the dorsal duct can be seen emerging from the minor papilla and coursing through the hyperechoic dorsal pancreas, which remains within the same, plus an identifiable point of crossing the CBD).; (II) From the stomach, the same finding can be demonstrated. (III) When seen from the descending duodenum, the main pancreatic duct is not identified and the ventral duct appears short and ending abruptly within the hypoechoic ventral part, giving rise to suspicion of pancreas divisum.

## 7. Diagnostic Accuracy for Diagnosis of Pancreas Divisum by EUS

Various criteria have been applied while evaluating cases of pancreas divisum with radial and linear EUS. In a study by Bhutani et al., investigators tried to obtain the stack sign in patients with pancreas divisum (*n* = 6) and compared them to the control (*n* = 30), and found a significantly lower percentage of patients with pancreas divisum having a demonstrable stack sign compared to the controls (33% vs. 83.3%, *p* = 0.04) [29]. A false positive stack sign was found in two patients. The accuracy and positive predictive value of the test was found to be 80%. In a study by Rana et al., two criteria were evaluated; viz., absence of the stack sign and the ability to trace the duct backwards from the papilla [28]. In this study, the authors found that the sensitivity and specificity of these two criteria were 50 and 97% and 100 and 96%, respectively. In the former study, EUS was done after diagnostic ERCP and the endosonologist was aware of the diagnosis (which can have led to bias); whereas, in the latter study, EUS was done prior to ERCP. Both studies used 7.5 MHz radial EUS for diagnosis.

Tandon et al. went beyond the mere absence of the stack sign to improve specificity of the EUS criteria [30]. They used two criteria for diagnosing pancreas divisum: (I) visualization of the bile duct and pancreatic duct entering the second part of the duodenum; (II) the pancreatic duct traversing the duodenal wall anteriorly and proximal to the bile duct. Using these criteria, the authors prospectively diagnosed two out of three cases of pancreas divisum without any false positive diagnosis.

Lai et al. studied 162 patients with linear EUS who, subsequently, also underwent ERCP [31]. They prospectively evaluated these patients using the trace-back sign and V–D transition sign, and the absence of any one of these signs was considered to be a diagnostic for pancreas divisum. They found the sensitivity, specificity, positive predictive value, negative predictive value, and accuracy of EUS for the diagnosis of pancreas divisum to be 95%, 97%, 86%, 99%, and 97%, respectively (Table 4).

## 8. Comparison of EUS vs. MRCP for Diagnosis of Pancreas Divisum

A systematic review and meta-analysis attempted to compare the diagnostic accuracies of MRCP, secretin-MRCP, and EUS for the diagnosis of pancreas divisum [32]. The authors included 10 studies with 856 patients evaluated with MRCP, 5 studies with 625 patients evaluated with secretin-MRCP, and 7 studies of 470 patients evaluated with EUS. They reported that EUS is more reliable than MRCP for the diagnosis of pancreas divisum but is inferior to secretin-MRCP. In terms of the effect of the diagnostic test, EUS performed better than MRCP but was found to be slightly inferior to secretin-MRCP (Table 5).

## 9. Role of Secretin-Enhanced EUS and EUS Elastography in Pancreas Divisum

Successful and accurate detection of pancreas divisum via EUS is influenced by multiple factors like expertise of the endosonologist and presence of a dilated pancreatic duct. Intravenous secretin has been used in EUS to make the pancreatic duct more prominent and thus improve the sensitivity of EUS for the diagnosis of pancreas divisum. In ERCP-based studies, it has been shown that secretin increases intraductal pressure within 1 min of administration, with peak effect on the pancreatic duct occurring within 3 to 5 min of administration [33]. Secretin has been shown to improve the visualization of the pancreatic duct and thus increases sensitivity for the diagnosis of pancreas divisum [34].

Secretin EUS has also been used to predict response to endoscopic minor papillotomy. Catalano et al. studied 40 patients with pancreatitis and pancreas divisum, and they reported that 81% patients with abnormal secretin EUS test had response to endotherapy whereas only 17% of patients with abnormal secretin EUS had failed endotherapy (*p* = 0.02). An abnormal secretin EUS test was defined as an increase in duct diameter by 1mm or more following secretin injection and the dilatation remaining at least for 1 mm throughout the 15 min test [35].

EUS elastography is an imaging modality that evaluates the stiffness of the tissues and has been shown to be useful in evaluation of chronic pancreatitis as well as in differentiating various pancreatic masses [36]. In pancreas divisum, role of EUS elastography is not well-defined. However, due to difference in echogenicity as well as stiffness of ventral and dorsal pancreas, EUS elastography can be useful in detecting ventral–dorsal transition which helps with the diagnosis of pancreas divisum [27].

## 10. Other Pancreatic Duct Anomalies: Ansa Pancreatica, Anomalous Pancreaticobiliary Union (APBU), and Role of EUS

Ansa pancreatica is one of the least common anomalies of the pancreatic duct, which is characterized by the absence of the accessory pancreatic duct and presence of an “S”-shaped branch arising from the main duct, which ultimately drains into the minor papilla [36]. It is different from the “meandering pancreatic duct”, where there is a looping of the distal part of the main duct which joins the patent duct of Santorini. It can be diagnosed with MRCP, ERCP, or EUS. There are only anecdotal reports of ansa pancreatica being diagnosed with EUS [37]. Due to the rarity of the disease, there is no data available to show the superiority of any diagnostic modality over other.

Anomalous pancreaticobiliary union (APBU) is another uncommon anomaly of the pancreatic duct with a prevalence of 1.5–3.2% [38]. It is characterized by an abnormal union of the pancreatic duct and common bile duct above the sphincter of Oddi, outside the duodenal wall. Multiple associations have been described in the literature, which include choledochal cysts, carcinoma of gall bladder, cholangiocarcinoma, chronic pancreatitis, and recurrent acute pancreatitis. There have been multiple proposed mechanisms for pancreatitis in APBU, which include biliopancreatic reflux, stenosis of the long common channel, obstruction of the channel by stones, epithelial hyperplasia, protein plug, and intraductal hypertension. MRCP and ERCP can identify APBU, although the latter is used with therapeutic intent only (pancreatic or biliary drainage). EUS has also been used to identify APBU, and the EUS criteria suggestive of APBU is the union of bile and the pancreatic duct outside the duodenal wall. Ko et al. reported that EUS detected 91% of patients with APBU via EUS [39]. Although EUS can potentially be used for the diagnosis of various rare congenital pancreatic anomalies, there is no clear consensus on preference. However, taking into consideration the high diagnostic yield of EUS in recurrent or chronic pancreatitis and acute pancreatitis patients of unclear etiology, EUS is likely to be helpful in establishing the diagnosis of these pancreatic duct malformations.

## 11. EUS in Annular Pancreas

Annular pancreas (AP) is a rare congenital anomaly that results from a failure of the ventral anlage to rotate clockwise with the duodenum during the embryogenesis, resulting in a thin band of pancreatic parenchyma encircling, either completely or partially, the second part of the duodenum near the major papilla [40]. Patients with annular pancreas may remain asymptomatic or present with pancreatitis, duodenal outlet obstruction, gastrointestinal bleeding, or abdominal pain alone. It is generally symptomatic in children, especially neonates, and in adults, it is usually an incidental finding on imaging.

It is an important differential diagnosis for descending duodenal obstruction and a contrast-enhanced CT can demonstrate the enhancing pancreatic tissue around the duodenum (Figure 5) [41]. ERCP is considered as the gold standard for the diagnosis of AP as it provides visualization of the descending duodenum and the pancreatic ductal system along with the major and minor papilla [42,43]. Duodenal narrowing is usually located just proximal to the major papilla, and the minor papilla is usually situated at the proximal rim of the annular ring. In the majority of patients, the duct of the annular pancreas communicates with the main pancreatic duct arising from the major papilla. However, ERCP via the major papilla will not be able to diagnose AP in patients where the annular duct opens into the minor papilla or directly into the duodenum [44]. Moreover, ERCP is technically difficult in patients with significant duodenal obstruction. The annular duct can also be detected on MRCP, where it commonly connects with the main pancreatic duct near the major papilla but also may drain into the dorsal duct near the minor papilla or directly drain into the duodenum [45]. MRCP, apart from being non-invasive and operator-independent, can also visualize pancreatic parenchyma around the duodenum.

EUS is not a primary imaging modality for the diagnosis of AP but is a useful investigational modality for elucidating the cause of extrinsic compression of the duodenum. Like MRCP, EUS can delineate both the pancreatic parenchyma and the pancreatic duct around the duodenum (Figure 6) and thus help with the accurate diagnosis of AP. Even in patients with duodenal obstruction, EUS can help with the diagnosis of AP by scanning from the duodenal bulb [44]. An added advantage of EUS is that it can help with the diagnosis of co-existent pancreatic pathologies like chronic pancreatitis and ductal calculi [46]. Papachristou et al. studied 5 patients with an AP with radial EUS and reported that CT could detect the AP in only 1 of 5 patients, whereas EUS could detect AP in all 5 patients [46]. EUS detected an encircling band of pancreatic tissue by approximately 360° in 3, 270° in 1, and 180° in 1 patient, respectively. Additionally, within this band of encircling tissue, the pancreatic duct was identified in 4 of 5 patients. The authors suggested following technical maneuvers during radial EUS to facilitate the identification of encircling pancreatic tissue:(a)use of glucagon to inhibit gut motility;(b)minimal balloon inflation to help avoid rapid uncontrolled scope withdrawal;(c)gentle scope deflection;(d)gentle forward pressure with an overinflated balloon into the narrowed region of the duodenum.

## 12. Conclusions

EUS provides high-resolution images of both the pancreatic parenchyma and the pancreatic duct and, therefore, is an ideal imaging modality for the diagnosis of various pancreatic duct anomalies. Studies have reported a high accuracy of EUS for the diagnosis of various pancreatic duct anomalies like pancreas divisum, APBU and AP. The use of secretin to improve the visualization of the pancreatic duct can improve the diagnostic capability of EUS to diagnose various congenital pancreatic duct anomalies. The ability to diagnose various co-existent pancreatic pathologies along with congenital duct anomalies is an added advantage of EUS. Being an operator-dependent imaging modality and the presence of multiple EUS criteria for the diagnosis of various pancreatic duct anomalies varying in sensitivity and specificity are important limitations of EUS. Further prospective comparative studies are required to establish accurate EUS diagnostic criteria for the diagnosis of various pancreatic duct anomalies and its exact role in the diagnostic algorithm for the evaluation of patients with suspected pancreatic duct anomalies.

## Figures and Tables

**Figure 1 diagnostics-13-03129-f001:**
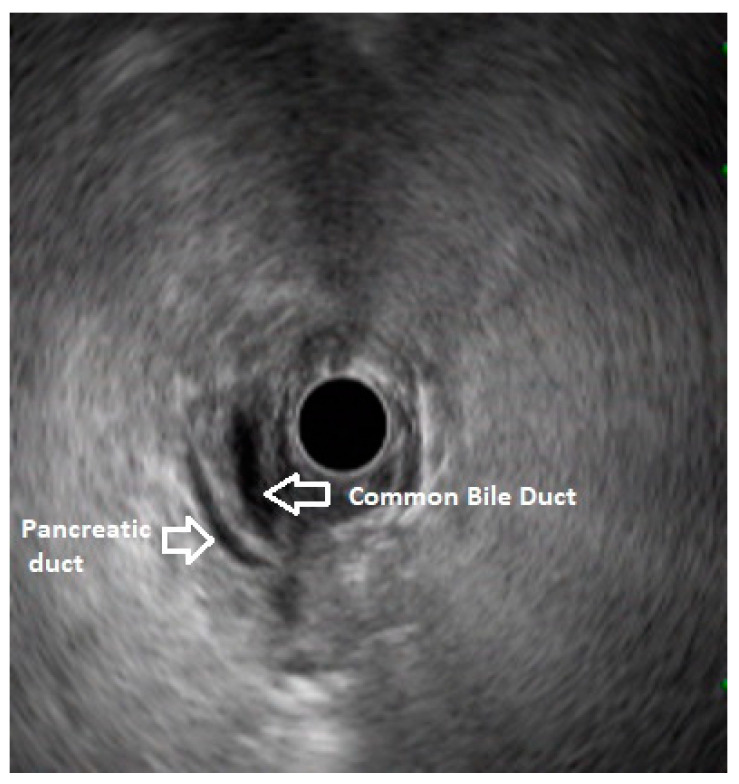
Radial EUS showing both ventral pancreatic duct and common bile duct stacked together. This finding rules out pancreas divisum.

**Figure 2 diagnostics-13-03129-f002:**
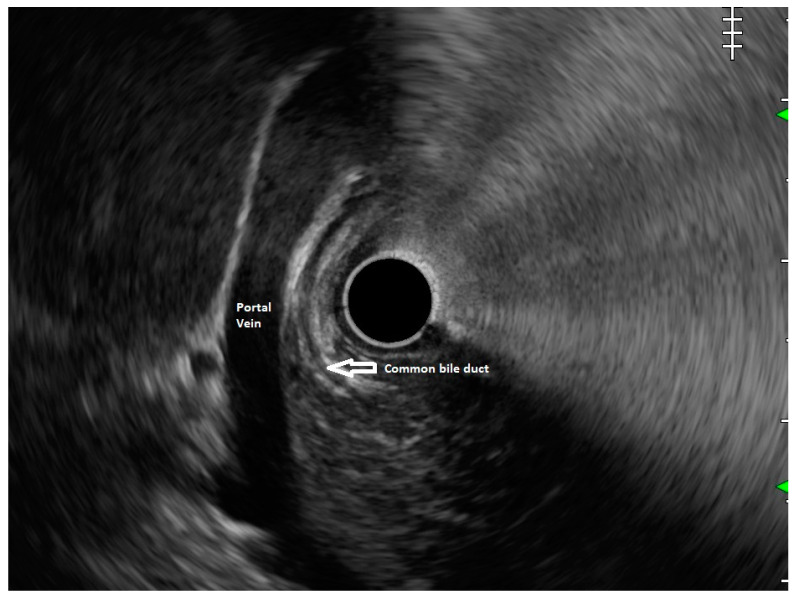
Radial EUS: Portal vein and common bile duct are seen but ventral pancreatic duct is not seen. This finding suggests diagnosis of pancreas divisum.

**Figure 3 diagnostics-13-03129-f003:**
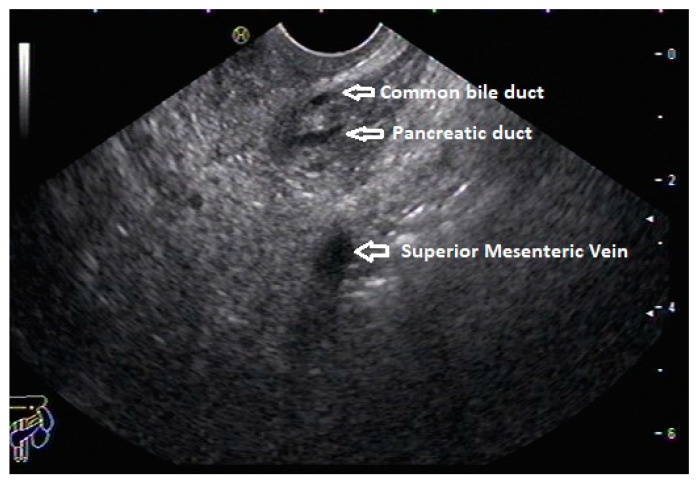
Stack sign on linear EUS.

**Figure 4 diagnostics-13-03129-f004:**
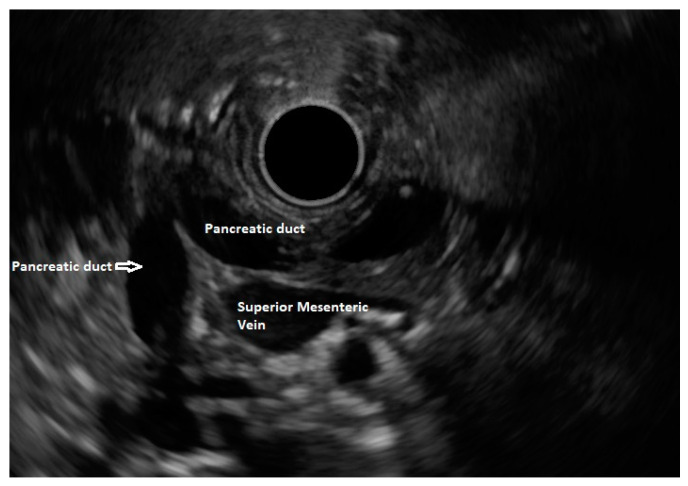
Pancreatic duct dipping towards major papilla excludes pancreas divisum.

**Figure 5 diagnostics-13-03129-f005:**
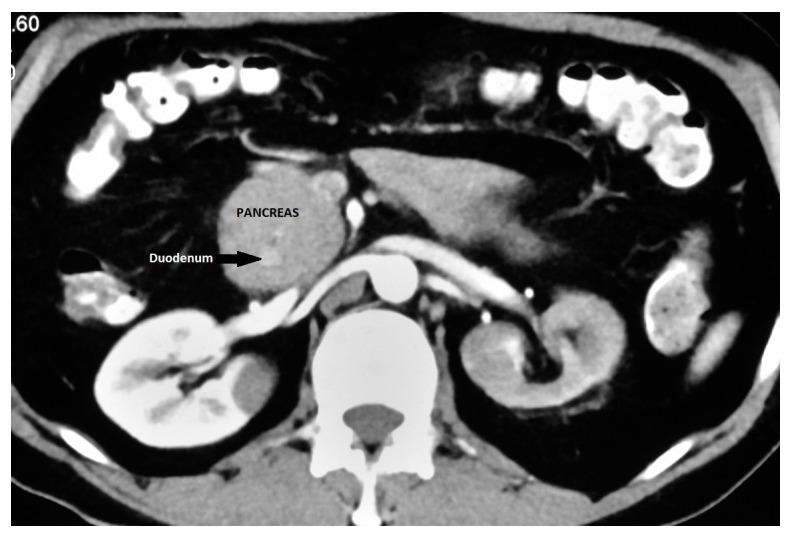
Annular pancreas: CECT showing pancreatic tissue encircling descending duodenum.

**Figure 6 diagnostics-13-03129-f006:**
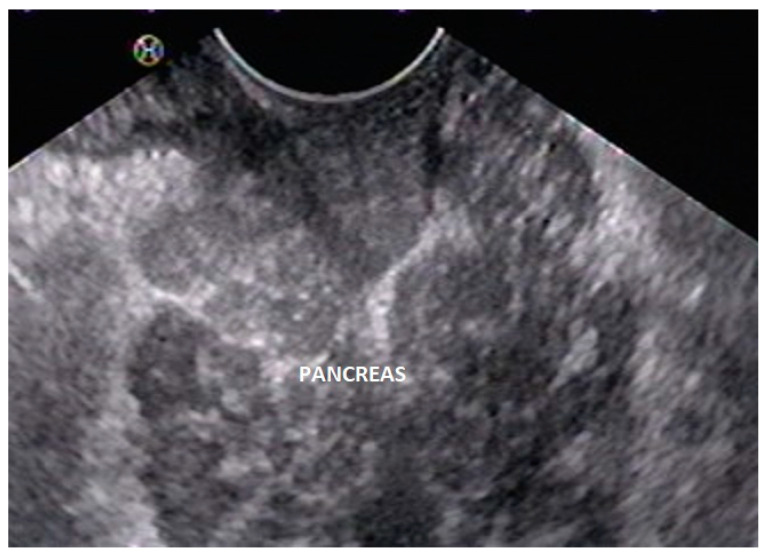
Linear EUS shows pancreatic tissue lateral to the duodenum.

**Table 1 diagnostics-13-03129-t001:** Types of pancreatic developmental anomalies and examples.

Type of Anomaly	Examples
Fusion anomalies	Pancreas divisum
Migration anomalies	Ectopic pancreas, annular pancreas
Duplication anomalies	Variation in number and forms

**Table 2 diagnostics-13-03129-t002:** Anatomical variations and resultant anomalies of the pancreatic duct.

Anatomic Variation	Type of Anomaly
Course of pancreatic duct	Descending
	Sigmoid
	Vertical
	Loop
Configuration of duct	Type 1: Bifid with dominant duct of Wirsung
	Type 2: Bifid with dominant duct of Santorini without divisum
	Type 3: Absent duct of Santorini
	Type 4: Pancreas divisum
	Type 5: Ansa pancreaticus
Anomalous union	Anomalous pancreaticobiliary union (APBU)
Pancreas divisum	Type 1: Total failure of fusion
	Type 2: Absent duct of Wirsung with dominant dorsal drainage
	Type 3: Presence of a small intercommunicating branch

**Table 3 diagnostics-13-03129-t003:** EUS features useful for the diagnosis of pancreas divisum.

Absence of stack sign Crossed duct sign Trace-back of the pancreatic duct Pancreatic duct not crossing ventral–dorsal transition Dipping sign absent	6. Absence or rudimentary ventral duct 7. Dilated duct of Santorini (more than duct of Wirsung) 8. Presence of rudimentary communicating duct between dorsal and ventral duct 9. Other findings like changes of (early) chronic pancreatitis in dorsal pancreas, Santorinocoele, stones, and microlithiasis in the dorsal duct

**Table 4 diagnostics-13-03129-t004:** Summary of studies using EUS for diagnosis of pancreas divisum.

Author	Sign	Number of Patients	Sensitivity	Specificity	Positive Predictive Value	Negative Predictive Value	Accuracy
Rana et al. [28]	Absent stack sign	20	50%	97%	73%	93%	91%
Rana et al. [28]	Inability to trace pancreatic duct from head to body	20	100%	96%	80%	100%	96%
Bhutani et al. [29]	Absent stack sign	30	44%	92%	66%	83%	80%
Lai et al. [31]	Trace-back sign and V–D transition sign	162	95%	97%	86%	99%	97%

**Table 5 diagnostics-13-03129-t005:** Summary of results of the systematic review and meta-analysis comparing the diagnostic accuracy of EUS with MRCP for the diagnosis of pancreas divisum [32].

Modality	Sensitivity %	Specificity %	HSROC	Diagnostic OR	Pooled LR^+^	Pooled LR^−^
EUS	85	97	0.97	167.89	26.80	0.16
MRCP	59	99	0.90	211.33	87.83	0.42
Secretin-MRCP	83	99	0.99	376.89	65.48	0.17

HSROC: hierarchical summary receiver operating characteristic curve; OR: odds ratio; LR: likelihood ratio.

## Data Availability

Not applicable.

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
