# Peer review of "Endoscopic Ultrasound in Pancreatic Duct Anomalies"

_diagnostics, 2023, doi:10.3390/diagnostics13193129_

Round 1

Reviewer 1 Report

In this paper, the authors try to review EUS technology for diagnosis of various pancreatic duct anomalies. I have the following comments for the authors: 

1. The abstract should rewrite. Main conclusions should be included in this section. 

2. What's the uniqueness or main contributions of this review paper? The authors should clarify this in the introduction section. 

3. What kind of review method did the authors use? 

4. Tables suggest to being added in each section to summarize the findings reported by references. 

Author Response

Reviewer 1:

We thank the respected reviewer and the editorial board for valuable comments that have helped us in improving our manuscript.

Response to the comments:

In this paper, the authors try to review EUS technology for diagnosis of various pancreatic duct anomalies. I have the following comments for the authors: 

Comment 1: The abstract should rewrite. Main conclusions should be included in this section. 

Response: As suggested, the conclusion has been modified.

2. What's the uniqueness or main contributions of this review paper? The authors should clarify this in the introduction section. 

Response: The same has been added in the revised manuscript in the introduction.

3. What kind of review method did the authors use? 

Response: The same has been added in the revised manuscript in the introduction

4. Tables suggest to being added in each section to summarize the findings reported by references. 

Response: The tables comparing various modalities is there in the manuscript as table no 4. Since, there are very few studies looking at the role of EUS in various ductal anomalies, we feel that addition of table will be duplication of the data presented in the text.  

Reviewer 2 Report

Reviewer's Comments on the Manuscript Entitled "The Value of EUS in Pancreatic Studies: An Evaluation of Anatomic Anomalies and Clinical Implications".

I have carefully reviewed the manuscript provided for revision and would like to express my gratitude for the opportunity to do so. The authors focus their attention on the role of Endoscopic Ultrasound (EUS) in the study of the pancreas, specifically examining its potential in detecting anatomical anomalies and their clinical implications. The central theme of the study is undeniably interesting; however, it does not introduce a particularly novel concept. The diagnostic potential of EUS is well-established in the literature. In my assessment, the primary weakness of this work lies in its lack of originality. Nevertheless, the manuscript is reasonably well-written, and the methodology appears sound. To enhance the overall appeal of the article, I recommend incorporating relevant case studies from the existing literature. The authors can draw inspiration from, for instance, the work published in doi.org/10.1016/j.amsu.2019.07.001, where three concurrent pancreatic tumors are documented. In such cases, EUS can indeed make a significant difference in diagnosis. Additionally, I suggest referencing the work in doi:10.1002/ccr3.2641, which explores the use of EUS in the study of cystic pancreatic lesions—a field of considerable interest for EUS applications. This would not only bolster the manuscript's international outlook but also bring a more contemporary feel to the reference list. In summary, while the manuscript provides valuable insights into the role of EUS in pancreatic studies, I recommend addressing its lack of novelty by incorporating relevant case studies and expanding the list of references to encompass a broader range of contemporary research. This approach would undoubtedly enhance the manuscript's overall appeal to the readership.

Can be accepted 

Author Response

Reviewer 2:

Comment: I have carefully reviewed the manuscript provided for revision and would like to express my gratitude for the opportunity to do so. The authors focus their attention on the role of Endoscopic Ultrasound (EUS) in the study of the pancreas, specifically examining its potential in detecting anatomical anomalies and their clinical implications. The central theme of the study is undeniably interesting; however, it does not introduce a particularly novel concept. The diagnostic potential of EUS is well-established in the literature. In my assessment, the primary weakness of this work lies in its lack of originality. Nevertheless, the manuscript is reasonably well-written, and the methodology appears sound. To enhance the overall appeal of the article, I recommend incorporating relevant case studies from the existing literature. The authors can draw inspiration from, for instance, the work published in doi.org/10.1016/j.amsu.2019.07.001, where three concurrent pancreatic tumors are documented. In such cases, EUS can indeed make a significant difference in diagnosis. Additionally, I suggest referencing the work in doi:10.1002/ccr3.2641, which explores the use of EUS in the study of cystic pancreatic lesions—a field of considerable interest for EUS applications. This would not only bolster the manuscript's international outlook but also bring a more contemporary feel to the reference list. In summary, while the manuscript provides valuable insights into the role of EUS in pancreatic studies, I recommend addressing its lack of novelty by incorporating relevant case studies and expanding the list of references to encompass a broader range of contemporary research. This approach would undoubtedly enhance the manuscript's overall appeal to the readership.

Response: We thank the respected reviewer for his valuable comments that have helped us in improving our manuscript. The suggested article  published in Annals of Medicine and Surgery talks about role of trans-abdominal ultrasound and therefore not discussed in the current review. Similarly, the other article published in clinical case reports talks about branch duct IPMN and Pan IN in IgG4 pancreatitis.

Round 2

Reviewer 1 Report

It seems the authors have not addressed my previous comments well, such as

1. Main conclusions should be included in this section. 

2. What's the uniqueness or main contributions of this review paper? 

4. Tables suggest to being added in each section to summarize the findings reported by references. 

Author Response

Comment 1: Main conclusions should be included in this section. 

Response: As suggested the conclusions have been added in the abstract as "

Along with the visualization of the course and configuration of the pancreatic duct, EUS can also visualize changes in the pancreatic parenchyma thereby helping in early diagnosis of any co-existent pancreatic disease. Absence of stack sign and crossed duct sign are important EUS features to diagnose pancreas divisum. EUS can also help in diagnosis of other congenital ductal anomalies like annular pancreas, ansa pancreatica, and anomalous pancreato-biliary union, although the published experience is limited." 2. What's the uniqueness or main contributions of this review paper?  Response: This has been mentioned in the introduction as "

The technical details as well as the accuracy of EUS in diagnosing congenital pancreatic anomalies has not been reviewed and therefore in this review article, we will focus on the role of EUS in diagnosis of various pancreatic duct anomalies. For this review, we performed an electronic search in PubMed using the following medical subject heading terms: Endoscopic ultrasound; congenital pancreatic anomalies; pancreas divisum; annular pancreas; ansa pancreatica and anomalous pancreato-biliary union. The search included all types of articles published in English Language until July 2023.  "

3.Tables suggest to being added in each section to summarize the findings reported by references. 

Response: Table 4 has been added in the revised manuscript. 

Reviewer 2 Report

See precedent review 

Author Response

(The authors gave the same response as above.)

Round 3

Reviewer 1 Report

I do not have further technical comments for the authors. 

Author Response

Thank you